# The Three Little Houses: A Comparative Study of Indoor and Ambient Temperatures in Three Low-Cost Housing Types in Gauteng and Mpumalanga, South Africa

**DOI:** 10.3390/ijerph17103524

**Published:** 2020-05-18

**Authors:** Bongokuhle Mabuya, Mary Scholes

**Affiliations:** School of Animal, Plant and Environmental Sciences, Faculty of Science, University of the Witwatersrand, Johannesburg 2000, South Africa; Mary.Scholes@wits.ac.za

**Keywords:** ambient temperatures, climate change, thermal comfort perceptions, low-cost housing, indoor temperatures, construction materials

## Abstract

Low-cost houses make up the majority of the homes in townships (racially segregated areas which are usually underdeveloped) in South Africa and there has been limited research on the indoor temperatures experienced by residents of these homes. As a developing nation the price and availability of construction materials, often takes precedence over the potential thermal efficiency of the house. Occupants of low-cost houses are particularly vulnerable to climatic changes which may increase the likelihood of exposure to extreme temperatures in South Africa. This study focused on the relationship between indoor and ambient temperature in two study areas namely; Kathorus in Gauteng and Wakkerstroom in Mpumalanga. Three housing types were included in the study (government funded apartheid era houses, government funded post-apartheid houses and informal houses (shacks)). Temperature data loggers were installed in each home, in each area, from June 2017 to July 2018. Ambient temperature data were collected for the period June 2017 to July 2018. The houses studied were built with different materials which affect their thermal efficiency. The study also included semi-structured interviews where occupant’s perspectives on housing could be surveyed. Household temperatures in Kathorus and Wakkerstroom, both in the warmer and colder months fluctuated substantially throughout the day. There was an 8 °C, 9 °C and 14 °C fluctuations in daily indoor temperatures of apartheid-era, post-apartheid and shacks houses, and daily outdoor fluctuations of 5–15 °C, with higher fluctuations measured in Wakkerstroom. Generally, ambient and indoor temperatures were correlated but showed high variability. Indoor data for the winter months were less well correlated. Data showed that residents are subjected to extreme temperatures and these are expected to increase. The householder’s perceptions of thermal comfort were often not related to indoor temperature readings but to behavioural changes including the use of warm clothes and wood burning stoves. The study’s findings suggest that a majority of low-cost houses are thermally inefficient especially for those built in the post-apartheid era and shacks. With these houses showing a clear link between ambient and indoor temperature fluctuations. The occupants of these homes are poor and vulnerable to health risks which could be exacerbated by temperature fluctuations. Small changes such as installation of ceilings and use of insulation could make a large difference in these houses.

## 1. Introduction

Africa’s, and in particular South Africa’s, major aim has been to reduce the occurrence of informal settlements and inadequate low-cost housing [1]. As a signatory to the global Sustainable Development Goals (SDGs), South Africa aligned its National Developmental Goals (NDPs) and human settlements plan to improving the living standards of approximately 1.5 million impoverished citizens, living in informal settlements and low-cost housing [2], thereby redressing the inequalities of the former apartheid regime. As recently as 2018, Statistics South Africa [3], 2017, has found that more than 20% of South Africans lived in state-subsidised Reconstruction and Development Project (RDP) low-cost housing and informal settlements. These homes are given to the people, at no cost, by the Government. A large percentage of these occupants reported being unhappy with the walls and roofs of their homes. South African temperatures are predicted to be increasing at twice the global average [4], making inhabitants of inadequately built low-cost houses even more vulnerable to not only climatic extremes but the overall increasing temperatures. Furthermore, few home improvements are made due to monetary constraints post-occupation making the inhabitants constantly at risk [5]. Many of the older occupants show co-morbidity related to Diabetes, Human Immunodeficiency Virus/Acquired Immune Deficiency Syndrome (HIV/AIDS), Tuberculosis (TB) and Hypertension and have low resistance to further risk. Ormandy and Ezratty [6] found that colder homes placed physiological stress on those people that spend the majority of their time indoors namely new-borns, the elderly and people with pre-existing health conditions. The newer homes are mostly electrified but cultural norms, in rural areas, like Wakkerstroom, requires that homes retain a coal-burning stove for cooking and space heat [7]. There has been extensive research on indoor pollution and health due to a buildup of carbon monoxide concentrations and high PM_10_ levels in these homes. According to Scorgie et al. [8], approximately 2000 children die annually as a result of respiratory infections caused by mostly indoor air pollution due to poor household ventilation. Earlier Studies, in Brazil, performed during warmer weather conditions by Gouveia et al. [9] identified the same vulnerable subgroupings, additionally including black and poorer people. The current paper directly speaks to these groups.

The last 60 years of research have provided South Africa with one of the most accurate and comprehensive climate records in all of Africa, however, not enough is known about the relationship between indoor temperatures in low-cost homes and ambient temperatures. This was studied in the three most common low-cost housing types in the country using temperature data for the warmer and colder months. The study was conducted in two areas with micro-climatic differences, a warmer area, Kathorus, located in Gauteng [10] and Wakkerstroom, located in Mpumalanga in a region known to experience very cold winter temperatures. These micro-climatic differences as well as, differences in the materials used to construct each type of housing, behaviour of the occupants and other household variables (e.g., number of windows), could inform us to their suitability as human dwellings. Tamerius et al. [11] found that the relationship between indoor and ambient temperatures was also dependent on socio-economic status, the indoor temperature of low-cost homes was more closely associated with ambient temperatures because low income homes did not have access to air-conditioners. Low-income homes use windows and doors for ventilation, which in turn results in notable correlations between indoor and ambient temperatures.

The Government is mandated with providing many low-cost houses at a minimum cost. Tenders are awarded to contractors often based on cost rather than reputation and this often leads to cost cutting practices including the use of below par construction and sanitation materials. For occupants, there is a negative social stigma associated with living in these homes which are perceived as being of a poor standard. A universal standard has been conditionally suggested by the World Health Organization (WHO) [12] which has widely recognised that housing cannot be viewed as “affordable or low-cost” if it compromises the occupants’ well-being and threatens their other basic human rights [13]. Unhealthy and unhappy [14] citizens show poor economic performance, thereby burdening an already overwhelmed health system with patients that cannot afford private healthcare. By building adequate low-cost houses a positive feedback loop will be created between the economy and the South African Public Health system.

In this paper, we addressed two related research questions:1)What is the extent of the relationship between indoor and ambient temperatures of the three housing types in each area, during warmer and colder months?2)Which factors, including housing type, location, behaviour and indoor temperatures, contribute most to an understanding of the perception of housing satisfaction?

## 2. Methods and Materials 

This study was conducted during June 2017 to July 2018 in two separate townships in South Africa, namely; Kathorus in Gauteng and Wakkerstroom in Mpumalanga. The two locations were chosen because they had different micro-climatic conditions despite being in neighbouring provinces. Both Kathorus and Wakkerstroom are located in the Highveld region of South Africa, the region is semi-arid and has highly seasonal rainfall (means of about 700–750 mm) and hot-wet summers with cold winters. The two townships had three similar low-cost housing types, namely; apartheid era “matchbox” houses, post-apartheid era “RDP” houses and informal “shack” houses (Figure 1, Figure 2, Figure 3, Figure 4, Figure 5, Figure 6, Figure 7 and Figure 8). All houses were electrified. The informal “shack” homes sampled were in the backyards of formal low-cost homes namely; matchbox or RDP homes. All houses had windows and doors, mostly with two doors opening to the outside but shacks had only one door. All matchbox homes (Figure 3 and Figure 7) had four small windows (two in front and two at the back), while shacks had between one and two windows. RDP houses in Kathorus (Figure 4) had two windows in the front and one at the back, all RDP houses in Wakkerstroom (Figure 6) had two front and back windows. The formal low-cost houses that were chosen were older than 20 years, in order to minimise building-age-based bias but shacks were often not older than 10 years. The houses were orientated true north, with the room in which most waking hours was spent facing north. Wall heights are a standard 2.3 m except for shacks, which varies based on the building materials used. 

### 2.1. Description of the Study Sites and the Three Types of Houses Sampled 

No housing plans are shown for the shacks (Figure 5 and Figure 8) they are invariably self-built and vary substantially. Shacks do not have any indoor plumbing, and often use materials such as curtains as bedroom partitions. The RDP homes like the matchbox homes were orientated to the north. Newer RDPs included indoor toilet facilities, while older RDP and matchbox houses had standalone outhouses, none of the homes had indoor bathroom facilities (Figure 1 and Figure 2). While the apartheid matchbox homes were supposed to be 53 m^2^ the houses studied were between 36–40 m^2^. Whilst the floor area was expected to be standard it did vary between 36–40 m^2^, with shacks being the most variable (20–30 m^2^). This is a limitation of the study.

### 2.2. Data Collection 

#### 2.2.1. Long Term Ambient Temperature Data

Hourly ambient temperature data were sourced from the South African Weather Services (SAWS) from June 2017 to July 2018. Kathorus’s data were sourced from Johannesburg INT WO station number: 0476399 0, coordinates −26.430: 28.234 and an altitude of 1695 masl. Wakkerstroom had no working weather station at the time of the study, hourly temperatures were calculated using an algorithm given by CIBSE [15], using daily temperature Tmax and Tmin readings supplied by the “The Wakkerstroom Bird Club”, which is a member of Birdlife Africa and it is required to keep weather data. The CIBSE method is highly regarded, however can produce a daily error of ±0.5 °C for any given month.

#### 2.2.2. Indoor Temperatures 

Thirty-six houses were chosen (18 at each site; six houses per housing type) using temperature monitoring devices (iButtons) and 28 devices were retrieved by the end of the study (8 buttons were lost by the occupants). The iButton has an accuracy ±1 °C, and sensitivity of 0.5 °C, the device can measure temperatures between the ranges of −40 °C to +85 °C. Community leaders (in both areas) identified the first few houses (to participate) and members of the community referred the rest using the snowball sampling technique [16]. The number of houses were chosen so that there was at least 3 replicates per housing type (in each area) to allow for variation between occupant behavior and building age. The temperature data loggers were placed in a common area (the main living room) in the houses (on top of a cupboard), away from any external heat sources including direct sunlight, logging temperatures at three-hourly intervals. The data loggers can only hold approximately 2,000 readings, they were placed in homes 01 June 2017 and retrieved 30 July 2018, they were replaced mid-way (12–15 March 2018) through the study. The iButtons were recovered from six matchbox, three RDPs and six shacks in Kathorus; and four matchbox, four RDPs and five shacks in Wakkerstroom. Comparisons were made using mean indoor temperatures per housing types in the two areas bearing the mind that the other variables in the study could not be controlled.

#### 2.2.3. Construction Materials

Accessing information from the Department of Housing and Human Settlements was difficult, so data on materials used (including brick, floor and ceiling type) per housing type were collected using expert interviews, occupant interviews as well as researchers’ observations.

#### 2.2.4. Occupants’ Behavioural Adaptations 

Information on appliances used such as refrigerators, types of stoves (coal, wood or electric), heaters and fans were recorded using researcher observations as well as occupant interviews. Information on clothing modifications, occupant behavior and time spent inside household were given by occupants. 

#### 2.2.5. Interviews

Semi-structured in-person questionnaires (in IsiZulu- primary language in both areas) were used to determine the occupant’s perception of their thermal comfort as well as the construction materials were conducted twice, during the months of July 2017 (colder months) and again February 2018 (warmer months). All respondents needed to be over 18 to participate, a total of 85 respondents participated in the study.

### 2.3. Data Analysis

The data were analysed using XLStat and RStudio. The monthly and hourly means of both indoor and ambient temperatures were analysed per location and housing type. Hourly temperature differences between housing types were calculated using a repeated-measured ANOVA and pairwise comparison between each site. Furthermore, a linear regression model (R^2^) and Pearson’s correlation (r) was used to establish the relationship between ambient and indoor temperatures. Participant’s thermal satisfaction responses were calculated using percentages and categorised per housing type and location. Categorical variables such as ceiling, floor, and roof type were given as percentages.

## 3. Results

### 3.1. Indoor and Ambient Temperature 

#### 3.1.1. Indoor Temperatures of the Three Housing Types in Kathorus and Wakkerstroom during Selected Times throughout the Day during Colder and Warmer Month

Figure 9 provides a visual representation of indoor temperatures at key times during the day (12 am, 6 am, 12 pm and 6 pm), of the three housing types in each area. The warmer months saw large daily indoor temperature fluctuations of all three housing types: matchbox houses (approximately 8 °C), RDP (9 °C) and shacks (approximately 14 °C) in both areas. There was a generally low variation in temperatures during the studied times throughout the warmer seasons, except for shacks. This variation was particularly high at 6 am when indoor temperatures of shacks were elevated, as a result of household activities such as cooking and heating which caused internal heat gains. The colder months showed relatively low daily indoor temperature fluctuations for matchbox houses in Kathorus (2.5 °C) and high daily temperature fluctuations for matchbox houses in Wakkerstroom (about 10 °C). In the colder months RDP households showed the greatest amount of daily indoor temperature fluctuations, approximately 11 °C in both areas. Shack temperatures in Kathorus showed lower than expected daily indoor temperature fluctuation (5.6 °C), while shack temperatures in Wakkerstroom predictably displayed a high variation (12.5 °C). The colder months displayed a high amount of temperature variation in all three housing types, across both areas during the studied times, particularly at 6 am (Figure 9). Temperature variation was the greatest in Kathorus for matchbox and shack homes (at 6 am), temperatures at this time were high for all sampled shack houses. Furthermore, in both Kathorus and Wakkerstroom, there was substantial temperature variation for RDP homes at 6am when the houses reached their lowest temperatures.

Shack temperatures were often highest early mornings (6 am) when ambient temperatures were low (Figure 10), this is likely as a result of behavioural adaptations such as early waking times and internal heat gains caused by cooking. Shack residents are unlikely to remain in their homes midday, either because of socialising or because these are common working/school hours, which could contribute to internal heat loss. Colder months display high temperature variation, particularly between 12:00 pm and 18:00 pm daily.

#### 3.1.2. Regression Model Comparisons of Indoor Temperatures of the Three Housing Types in Kathorus and Wakkerstroom as a Function of Outdoor Temperature

Table 1 provides a Pearson’s correlation with p-values of the indoor and outdoor temperatures experienced within each housing type for warmer and colder months. During the warmer months both areas had indoor and outdoor temperatures which were statistically correlated for matchbox and RDP houses ((r = 0.74, *p* < 0.0001), respectively). This trend was also present in Wakkerstroom for shacks (r = 0.6, *p* < 0.05), however Kathorus’ indoor shack temperatures were not correlated with outdoor ones (r = 0.05, *p* > 0.05). The differences in indoor and ambient correlation were apparent during the colder months in Kathorus. During the colder months there was no correlation (*p* > 0.05) in indoor and ambient temperatures of two housing types in Kathorus (matchbox and shacks, r = 0.15 and r = 0.03 (respectively)), while RDP homes in the same area had a significant correlation (r = 0.75, *p* < 0.0001). All three housing types in Wakkerstroom had indoor temperatures which were significantly correlated with ambient temperatures (Table 1). 

The regression models presented below in Figure 11, shows the graphical relationship between indoor and ambient temperatures for all three housing types. The regression models were created for homes with and without ceilings to first create to gain understanding on the average indoor temperatures of any randomly selected household in both areas. Then the apparent independent effect of ceilings on indoor temperatures was then analysed separately.

During warmer months indoor temperatures of shacks in Kathorus, as well as matchbox and RDP houses of both areas displayed a relatively strong association with ambient temperatures (Figure 11). 

However, Wakkerstroom shacks deviated from this trend (R^2^ = 0.221), suggesting that indoor temperatures were highly variable. Colder month indoor temperatures of matchbox and shacks homes in Kathorus showed a weak relationship with ambient temperatures (Figure 12). However, RDPs in Kathorus and all three housing types in Wakkerstroom displayed a relatively strong indoor and ambient temperature relationship. Suggesting that outdoor temperatures were a good indicator of indoor ones. 

Regression are presented in Figure 13 for matchbox and RDP (n = 4) with ceilings and those without ceilings (n = 8) for comparative purposed. None of the homes in Kathorus had ceilings, floor plans of matchbox and RDP homes do not include ceilings. Ceilings were installed post-occupation and so their occurrence was limited to affordability. 

Figure 13 suggests that ceilings affect the thermal properties of homes, when ceilings are absent indoor temperatures were strongly related to ambient ones in both matchbox and RDP houses (R^2^ = 0.8153 and R^2^ = 0.9362 (respectively)). The houses with ceilings displayed less correlation with ambient temperatures, they were about 25% (matchbox) to 36% (RDP) less correlated when compared to those without ceilings. The sample size was very small (n = 4) and even though it gives some information on the effect of ceilings it can only be used speculatively. 

#### 3.1.3. Seasonal Indoor Temperatures Comparison between Three Housing Types in Kathorus and Wakkerstroom

During warmer months in both Kathorus and Wakkerstroom, average seasonal indoor temperatures of matchbox and RDP houses were not statistically different (*p* > 0.05) (Figure 14). However, during colder months the box plot in Figure 12 indicates that in both areas matchbox and RDP houses had statistically different (*p* < 0.05; r = 0.65) indoor temperatures. In the warmer months differences between indoor temperatures of RDPs and shacks were not statistically significant, implying that they had similar indoor temperatures. In colder months, only shacks in Kathorus had indoor temperatures which were moderately similar to those of RDP houses (Figure 14). For both colder and warmer months matchbox houses had indoor temperatures which were statistically different to shack indoor temperatures of the same area, consistently having daily mean temperatures higher than the other two housing types. All three housing types in Kathorus generally had temperatures which were higher than in those in Wakkerstroom, except for matchbox houses in Wakkerstroom during colder months. During the colder months the lowest mean daily temperatures were measured in shacks of Wakkerstroom (reaching lows of 3 °C), warmest daily temperatures were measured in shacks of Kathorus (reaching highs of 40 °C). Daily mean indoor temperatures of RDP houses in Kathorus during warmer months showed the least seasonal variation, during the warmer months (Figure 12). An ANOVA (represented by a box-plot) was performed on the data to determine the variance among and between samples, there was little difference within the sample housing types but significant (*p* < 0.05) differences between the three housing types (Figure 14). On average shacks had the widest range of temperatures (Figure 14).

### 3.2. Construction Material and Self-reported Information from Housing Occupants

Access to standardized construction methods and materials of low-cost houses in South Africa is restricted, information was gathered by means of interviews, which included occupant responses, experts on South African housing and the researchers own observations (this is especially true for informal houses). Table 2 presents a summary of the construction materials including floor material, ceiling material and roofing materials used to construct each type of low-cost house, as well as mean age of dwelling. No houses in Kathorus had ceilings present, while less than 30% of the houses sampled in Wakkerstroom had ceilings (across all three housing types). The types of ceilings used in Wakkerstroom differed, from the commonly used rhino-board type ceilings (polypropylene sheets) to the traditional wood ceilings and finally, self-constructed cardboard ceilings (found in a single shack). Concrete was used in the flooring for all housing types. 

There was no insulation materials used in-between the walls of the all houses, but those houses with ceilings had ceiling insulation. In the shacks of both areas vinyl was used to cover the raw concrete. In Kathorus, most matchbox houses (n = 5) had tiled floors, while only one matchbox house (n = 1) in Wakkerstroom had this feature.

Matchbox houses in both Kathorus and Wakkerstroom were approximately 60 years old, while RDP houses in both areas were approximately between 21 to 24 years old (Table 2). Shacks in Kathorus were no older than 4 years while in Wakkerstroom were between the ranges of 0.5 to 18 years. Shacks in Wakkerstroom were more easily dismantled as they were usually without a concrete foundation, while in Kathorus backyard shacks were secured with a slab of concrete and represented a more long term accommodation. Wood or coal stoves were used in Wakkerstroom for cooking and heating whilst electric stoves and electric heaters were used in Kathorus (Table 3). All households used refrigerators for food storage. On average Wakkertsoom matchbox houses had the highest number of occupants (n = 6) followed by RDP houses (n = 5). This does add a confounding factor to the analysis of the results but it was a variable that was impossible to eliminate.

#### Residents Thermal Comfort Perceptions 

A five-point rating scale was used (Figure 15) with responses ranging from too hot to too cold. The semi-structured in-person interview was performed twice, once during the warmer months and once during the colder months. In Kathorus (Figure 13), during warmer months, 67% of matchbox and shack (respectively) dwellers felt their homes were too hot, and only 33% felt thermally satisfied. All RDP dwellers in Kathorus, reported feeling that during warmer months their houses were slightly warm (but not intolerable). Similarly, in Wakkerstroom 67% of matchbox and shacks dwellers (respectively) communicated feeling their homes were too hot during the warmer months. RDP Households in Wakkerstroom had an even percentage (50%) of dwellers that felt their homes were too hot and those that reported feeling thermally satisfied. 

During the colder months (Figure 15) in Kathorus, a majority (67%) of matchbox house dwellers felt thermally comfortable, and 16% of residents felt the house was slightly cool and another 16% felt it was too cold. In Kathorus all sampled RDP dwellers felt that temperatures inside their homes were slightly cool, while all shack dwellers felt that their homes were too cold and offered little to no thermal comfort. In Wakkerstroom, during colder months, a large majority of dwellers (across housing types) felt their homes were too cold. For both matchbox and RDP houses, 83% of the occupants reported having houses that were too cold. Interestingly, the homes that reported feeling thermally satisfied were the homes with ceilings. In Wakkerstroom during colder months all shack dwellers communicated feeling thermally dissatisfied and that their homes were too cold.

## 4. Discussion

The discussion will focus on three areas. Firstly the relationship between indoor and ambient temperatures, secondly a discussion on construction materials and temperatures and thirdly a section on the perceptions of the occupants and some speculation on potential health problems. The two locations used in this study were primarily chosen because of the differences in ambient temperatures especially for winter in Wakkerstroom. The interpretation and discussion of the data based purely on climatic criteria are difficult because the behaviour and wealth status of the people in the two areas is very different. It appears best to examine the effects of temperature and behaviour across housing type within an area rather than to emphasize the comparisons between the two areas except where the evidence is convincing. Kathorus can be described as a peri-urban township as it is only 35 km’s east of Johannesburg and is surrounded by an industrial area. Wakkerstroom is 270 km’s south east of Johannesburg and is located in a rural farming community. Unemployment in South Africa is 29% and it is high in both the areas studied but much higher in Wakkerstroom. During the apartheid era, housing was made available, on a subsidised basis, to poor people as part of a Government Housing Plan. Since democracy in 1994, the African National Congress is committed to providing housing to the poor at a very large scale. 

### 4.1. Relationship between Indoor Temperatures of Low-Cost Houses and Ambient Temperatures

During the same season, indoor thermal conditions were different for each housing type at the two locations (Figure 11 and Figure 12); but similar patterns did emerge. In the warmer months, barring shacks, all other housing types’ indoor temperatures were strongly associated and correlated with ambient temperatures. This study found that when dwellings were made from metal, not only was correlation low in the warmer months, but temperatures were often several degrees higher than ambient temperatures. The houses studied showed high daily temperature fluctuations, exceeding permissible ranges (of about 2–3 °C) given by ASHARE Standard 55 [17]. Shacks had the lowest thermal mass, and the highest daily thermal fluctuations (14 °C to 15 °C), RDP and matchbox houses (8 °C, respectively) also had notable daily thermal fluctuations. Numerous international studies [9,18,19,20] have shown the association between highly elevated temperatures and high thermal fluctuations which have been linked with numerous health issues particularly cardiovascular ones and mortality.

Shack residents in Kathorus and Wakkerstroom were particularly vulnerable during the warmer months, due to exceedingly high indoor temperatures. Maximum temperatures inside shacks were found to be particularly high, reaching 33 °C in Wakkertstroom and 42 °C in Kathorus. The mean indoor temperatures measured in Kathorus shacks were consistently the highest (during warmer months) of any other housing type or site. According to Bröde et al. [21], and the Universal Thermal Climate Index (UTCI) ambient temperatures above 35 °C, placed substantial physiological strain on the body, and could be life-threatening. Supposing that the same temperature threshold applied (indoor temperatures above 35 °C) then temperatures in Kathorus are particularly noteworthy. The severity of risk however, is largely age-related, very young and elderly people are at particular risk, this group accounted for about 30% of the shack dwellers in Kathorus. The other two types of housing (in both locations) had daily temperatures which fell within or near accepted South African thermal comfort standards (18–25 °C) suggested by Makaka and Meyer [22]. According to WHO [12] as many as 32 studies in temperate regions were able to show a relationship between indoor and outdoor temperature, particularly when temperatures were greater than 20 °C, which corresponds to findings in this study. Even though this study used crude methods to show a relationship between indoor and ambient temperatures it had similar findings to studies which used more advanced methods suggested in ASHRAE 55. This is possibly due to the weak relationship between indoor and outdoor Relative Humidity (RH) [23], making RH a weak predictor of thermal comfort particularly in hotter regions.

During the winter months, in both Kathorus and Wakkerstroom the correlation of indoor temperatures with ambient became much weaker. However, what did emerge for the Wakkerstroom data set was that the occupants were very vulnerable to low temperature extremes especially in RDP and shack houses that have low thermal mass and high thermal conductivity, and quickly responded to ambient changes. This places RDP and shack dwellers (particularly in Wakkerstroom) at increased risk of cold related illnesses. The houses sampled did not seem to provide protection for occupants especially during temperature extremes. The low-cost houses studied here displayed significant levels of temperature variation between them, and between seasons, which supports the findings of similar studies conducted in the Gauteng and Eastern Cape areas, in South Africa by Naicker et al. [24] and Makaka and Meyer [22]. The findings in this study point to high indoor temperature variation during colder months. A similar study conducted by Naicker et al. [23] in the Johannesburg area was able to demonstrate increased daily temperature variability as well as high daily fluctuations in RDP as well as informal houses. This study was able to expand on these findings showing that all three houses showed increased daily temperature variability as well as high daily indoor fluctuations. Variability was to a great extent a function of season, location and housing type. Work by Nguyen et al., [23] in Boston, in the United States of America, supported the concept of differing impacts that either warmer or colder weather had on indoor temperature variability. 

### 4.2. Choice of Construction Materials and its Effect on Indoor Temperatures 

The choice of construction materials should largely be climate dependent. This is true for developed countries, but because South Africa is a developing nation, the price and availability of building materials often takes precedence in what is used to build low-cost houses. Ideally materials would have an optimized balance between good insulation and high thermal mass. But this is often not the case. During the warmer month’s homes built with clay brick (matchbox houses in both areas) had ambient temperature which were relatively strongly associated with indoor temperatures. These findings support studies by Maoto and Worku [25] that suggest inadequate construction (even with adequate construction materials) lead to defects that display themselves in inadequate thermal regulation. Houses in the Wakkerstroom had on average more cracks (researcher observed) which is possibly the reason this strong association exists. Even so during the colder month’s matchbox houses (in both areas) often retained heat better and had on average higher daily indoor temperatures than the other two housing types. Supporting research by Naicker et al. [24] and Makaka and Meyer [22] suggesting clay brick as being better than cement brick or metal sheets (used to construct the other two housing choices). Makaka and Meyer [21] stated that these bricks were highly thermally conductive (0.8 Watts per meter-Kelvin), less heavy, had a lower thermal mass and also had reduced insulation capacity than conventional stock bricks. The use of cement brick is not a poor choice of material, provided good plastering and insulation material is used. The plaster just serves the purpose of filling the cracks and partially sealing the brick surfaces. The RDP houses sampled were often not plastered or insulated which explains the relatively insufficient insulation offered. The choice and ratio of each constituent building material (for example, what proportion of cement, to river sand to ash should be used) is largely contractor dependent. In both areas, during both seasons’ the houses were not thermally ideal to inhabit but houses in Kathorus were slightly better than those in Wakkerstroom. 

Where ceilings were present, indoor temperatures of matchbox and RDP houses were about 25% and 36% (respectively) less associated with ambient temperatures, compared to similar households in the same area. Previous and existing low-cost housing regulations had/have not required ceilings, insulation or good plastering in order for building projects to be approved. Adding ceilings (and insulation) could greatly improve the thermal quality of all housing types, even though the sample size was extremely small in this study this is an area worth pursuing and presenting to the building council for inclusion in the policy documents. The houses studied did not have wall insulation which negatively impacted their thermal efficiency. Insulation is an important factor in creating a thermal envelope (barrier protecting the house from unwanted heat loss to the environment) around houses, houses without insulation expose occupants to increased vulnerability to non-communicable diseases, ceilings (and wall insulating materials) were not a common feature in the studied areas. It has long been accepted that asbestos and isocyanate ceilings and roofs place considerable stress on human health, and even though they are good at offering insulation they pose a huge threat to human health [12]. The orientation of the homes was north facing but the planning and design did not consider the large number of sunshine hours that the whole of South Africa receives. Some of the homes are solar geyser installed but none make use of solar panels for lighting or cooking.

The Government is well intentioned in terms of providing homes rapidly to the poor but there is a significant amount of corruption and the issuing of title deeds and home-ownership papers has been slow, and without ownership residents are often not motivated to make improvements to their houses. In the case of RDPs making alterations to homes means that occupants need to seek approval from the offices which issues said homes, which is often cumbersome. Adding to that, low-cost houses are inhabited by the poorest members of society who often cannot afford to add these structures post-occupation. The government cites monetary constraints, increased demand and inadequate capacity (municipal administrative issues) as reasons for issuing substandard housing.

### 4.3. Perception of Thermal Comfort and Satisfaction and Human Health

Low-cost houses are offered by the South African government to families who otherwise couldn’t afford to own property. According to Jacobs [13] low income groups are more likely to be exposed to the risks associated with unhealthy housing because they are more likely to live in unsuitable housing which is all they can afford. As anticipated the sampled population living in formal low-cost houses had the greatest amount of thermal satisfaction. Participants in this study particularly those in RDP homes reported having greater thermal satisfaction than those in shacks. An unanticipated finding was between matchbox and shack occupants, they reported similar levels of housing satisfaction during the warmer months even when daily temperature readings did not appear to support tenants’ perceptions. The extent of indoor temperature variability gives an indication of thermal comfort experienced by occupants. Increased indoor temperature variability typically relates to decreased thermal comfort (as the house does not maintain constant thermal conditions) [23]. Occupants should be most vulnerable when the ambient temperatures are at their lowest, but this is confounded by a number of other factors e.g., access to blankets, warm clothes and space heaters. As stated earlier, homes in Wakkerstroom, even though they have electricity, all have coal or wood burning stoves in their home. These are often lit early in the day to prepare breakfast and in winter months are kept burning gently through the day and then stoked in the late afternoon for preparing dinner and for warming the house during the night. Matchbox occupants that were more dissatisfied with their household temperatures during warmer months, lived in older homes with many people (range of between 1 and 9) which perhaps influenced their perceptions. However, RDP residents in general were satisfied with their newer homes during warmer months and reported having a higher standard of life. For shack, RDP and matchbox dwellers housing satisfaction and by extension thermal satisfaction seemed to be greatly associated with perceived wealth status. One RDP resident reported “At least I don’t live in a shack” another commented on how old matchbox houses were. Suggesting that thermal satisfaction was as much a physiological phenomenon as a socio-economic and psychological. Building energy efficient houses is important as hot and cold temperature extremes are increasing in frequency, the occurrence of non-communicable diseases such as asthmas, mold related allergies and other respiratory diseases could also increase [12].

South Africans, regardless of wealth status, are usually awake from 5 or 6am as work starts in the cities at 8 am, Schools at 7:15 am and farming activities at dawn. Poorer people have to get up considerably earlier as public transport in South Africa is poor. Owning a house in South Africa is an important status symbol coupled with this is a large migrant population which moves from rural areas to urban areas in the search for employment in small spatial steps. These people may move 150 kms in a year slowly heading for the bigger cities, they often occupy shacks along their journey spending 6–18 months in any one location. Their daily behaviour is very different to those people living in matchbox and RDP houses. Occasionally, families may occupy shacks but it is more likely to be young men who are unemployed and leave their homes very early in the morning to find work, especially in the Wakkerstroom area, where the work is mostly farm based. Shacks may have high temperatures at 6 am because of the occupants behaviour but is was an unexpected finding. In some cases, shack dwellers find employment in an area and then move up the social status ladder to be able to move into a matchbox or RDP house.

Colder weather temperature perception and standard indoor temperature conditions [17,22] and the actual measured indoor temperature readings were mostly aligned, suggesting that during colder weather tenants were acutely aware of their thermal environments. However, again during the colder months RDP dwellers (particularly in Kathorus) expressed the highest thermal satisfaction, which did not support thermal readings. Suggesting for RDP dwellers who are first time home owners housing satisfaction is not automatically associated with the actual thermal conditions experienced by occupants. 

IButtons were placed in the living rooms of formal low-cost homes and often in the only room in a shack. A single button was a limitation of the study but measuring more houses would have been better than placing multiple buttons in fewer houses. Occupants had thermal comfort perceptions which were often unrelated to measured indoor thermal readings (matchbox dwellers during warmer months). Environmental factors (e.g., humidity and drafts in the house) and socio-economic factors (such as wealth status, number and type of warm clothes and room warming devices), while not within the scope of this research, could perhaps offer some additional insight in future studies. Another confounding factor in this study was the very different ages of the homes and the varying sizes, it is difficult to draw substantive conclusions when there are so many other factors. 

In 2018 the WHO [12] released location specific housing and health guidelines which aimed to persuade policymakers to reduce the health risks associated with inadequate housing creating “health-promoting environments”. Appropriate housing is critical to health and well-being, a major short coming of this study is the lack of health data associated with these communities. Poorer citizens usually do not have access the medical aid systems and private health facilities. However, local health clinics provide essential medical services to millions of people on an annual basis. The legacy of apartheid has meant that access to data on health care is very difficult to obtain. This is mostly related to the stigma associated with immunocompromised people living with diseases such as HIV and respiratory infections such as TB. Clinic staff are only required to keep information on age, gender and a general diagnosis. The most common diagnoses in areas where there is either indoor or outdoor pollution are bronchitis, asthma and upper respiratory conditions. These ailments are ubiquitous amongst poorer people and it would be very difficult to link these specifically to changing environmental conditions unless one had access to an extremely large spatial and temporal data set. 

## 5. Conclusions

The study had a few challenges such as a low sample size and lost or missing temperature monitoring devices. Another limitation is not measuring air velocity or RH, which could give a deeper understanding of thermal comfort within the houses. However, clear trends could be seen. During warmer months shacks had indoor temperatures which were exceedingly higher than any other housing type in both areas. During colder months RDP and shack temperatures in both areas were highly variable and insufficient at maintaining internal thermal conditions throughout the day. A lot of valuable information was gained from interviews, perception around thermal satisfaction and suggested guidelines on acceptable indoor temperatures often did not coincide. Confounding factors such as wealth status, perceived control over their environment seemed to greatly impact occupant’s thermal satisfaction. 

## Figures and Tables

**Figure 1 ijerph-17-03524-f001:**
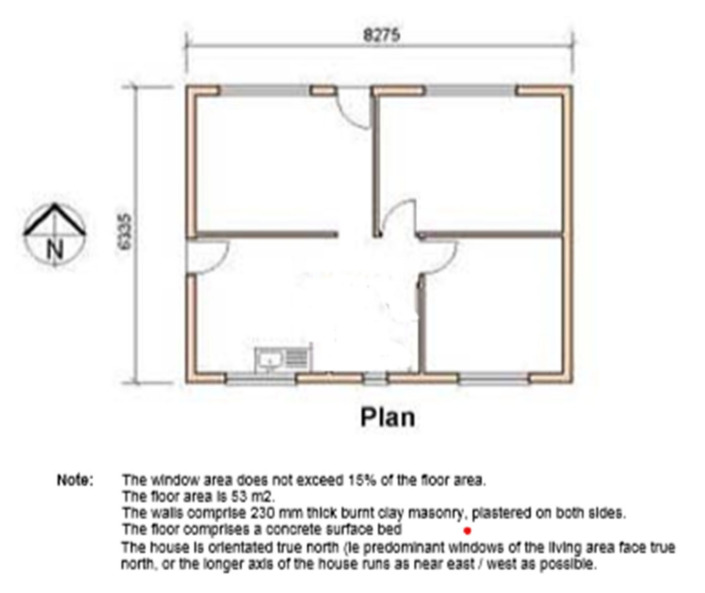
Floor plan of a common South African matchbox low-cost house dimensions slightly vary between locations however, main wall material (clay) remains constant. The average U-value of these homes is 42.56 W/m^2^ °K. (sourced from: http://www.nhbrc.org.za/files/technical_docs/Design_and_Construction_of_Houses.pdf).

**Figure 2 ijerph-17-03524-f002:**
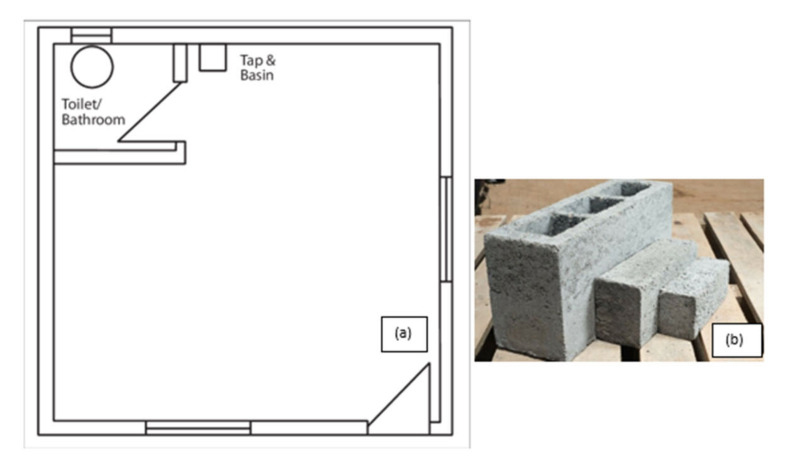
Floor plan of a common South African RDP home (**a**); in the last approximately 20 years RDP homes now include wall partitions creating two bedrooms, walls are made from cement brick (**b**), older homes are made with the more compact (150 mm × 100 mm) brick and newer homes (built after 1999) with the larger (200 mm × 400 mm) brick. The larger cement brick commonly is not filled with any insulating material. The homes have a U-value of 49.37 W/m^2^ °K (both old and recent RDPs). The RDPs that were sampled have a comparable surface area to apartheid era matchbox houses (sourced from: Moolla et al. [14]).

**Figure 3 ijerph-17-03524-f003:**
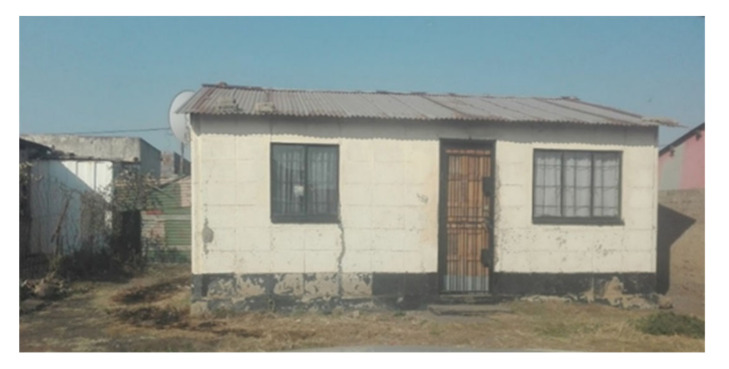
An archetypal Kathorus apartheid era- matchbox- house constructed in the early 1960s, housing between one to six occupants. The houses were typically 40 m^2^ in area, and were constructed with low-grade clay brick. The houses either had corrugated or asbestos roofing, and four windows (two in the front and two at the back).

**Figure 4 ijerph-17-03524-f004:**
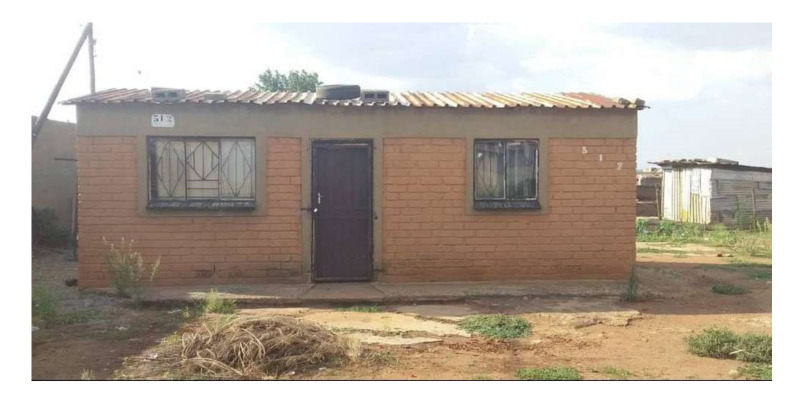
A distinctive RDP house in the Kathorus area, built in the mid- to- late 90s, housing between one to five occupants, with an area of about 36–40 m^2^. The houses were made with cement bricks. The house had asbestos roofs and three windows (at two windows front and one at the back).

**Figure 5 ijerph-17-03524-f005:**
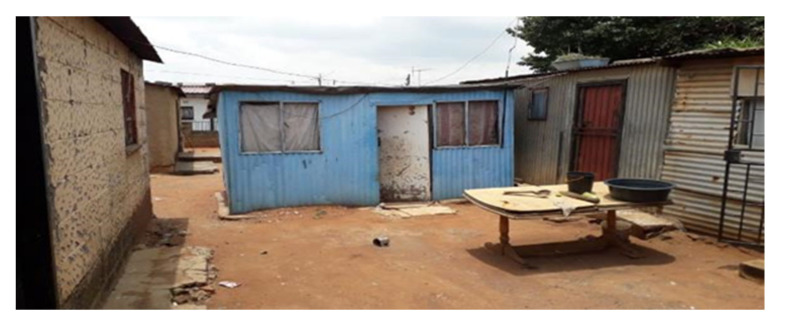
A typical backyard shack in the Kathorus area, shacks are usually no older than 10 years. Each backyard shack houses between one to three occupants. Shacks are usually made with corrugated iron and any scavenged materials. The number of windows was not standard, they were between 2 and 3. Shacks generally had a floor area of about 20–25 m^2^.

**Figure 6 ijerph-17-03524-f006:**
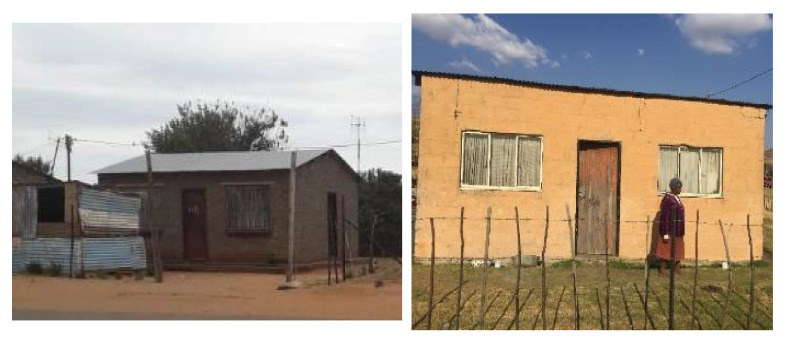
Examples of typical RDP houses in Wakkerstroom; the houses were constructed in late 90’s to early 2000s with maxi brick and corrugated iron roofs (left), and the house at the right is an example of earlier (1994) RDP constructions with asbestos roofing and only one door at the front of the home. The houses had similar dimensions to those found in Kathorus.

**Figure 7 ijerph-17-03524-f007:**
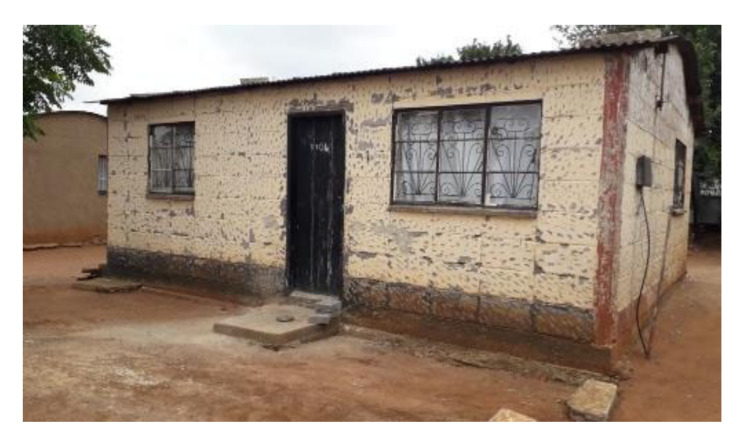
Apartheid era (matchbox) houses found in eSizameleni (Wakkerstroom), with a surface area of 40 m^2^. The houses had either asbestos or corrugated iron roofing with 2 doors and 3 -4 windows.

**Figure 8 ijerph-17-03524-f008:**
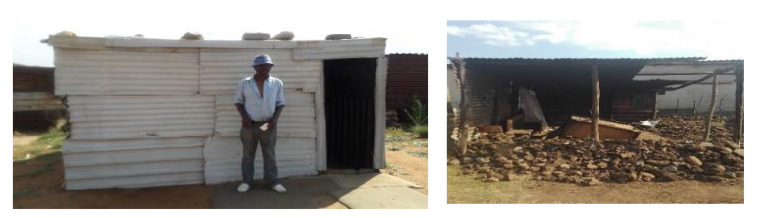
A typical backyard shack in Wakkerstoom (left), this housing type is not seen as a permanent dwelling, as evidenced by occupants dismantling it shortly after receiving an RDP from the government (right). The homes were occupied by one to three occupants. The homes were constructed with mainly corrugated iron, mud blocks as well as other scavenged or cheap materials. The homes generally had a floor area of 20–30 m^2^.

**Figure 9 ijerph-17-03524-f009:**
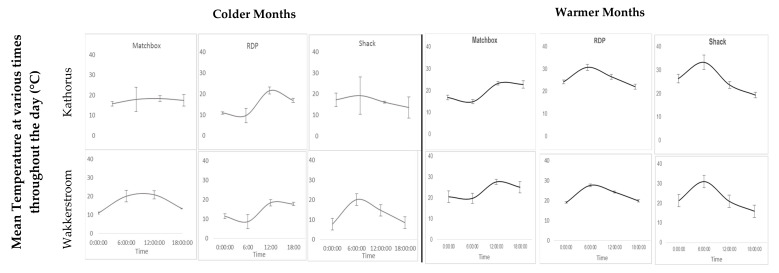
Mean indoor temperature readings at selected time throughout the day of the three housing types, during colder and warmer months in the Kathorus (top row) and Wakkerstroom areas (bottom row), error bars indicate daily mean temperature variance (n=16).

**Figure 10 ijerph-17-03524-f010:**
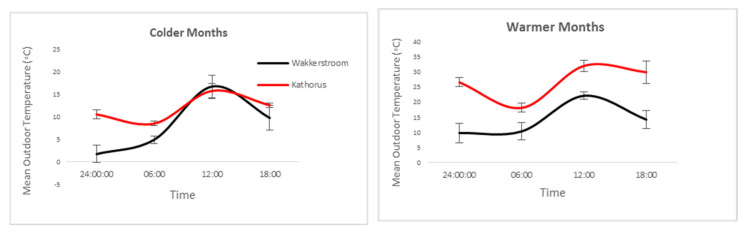
Mean ambient temperature readings (n = 4) at selected times throughout the day during colder (left) and warmer months (right) throughout the day in Kathorus and Wakkerstroom, the error bars indicate daily mean temperature variance (throughout the sampled period). Kathorus had ambient temperatures which were higher than Wakkerstroom, this is especially evident during the warmer months.

**Figure 11 ijerph-17-03524-f011:**
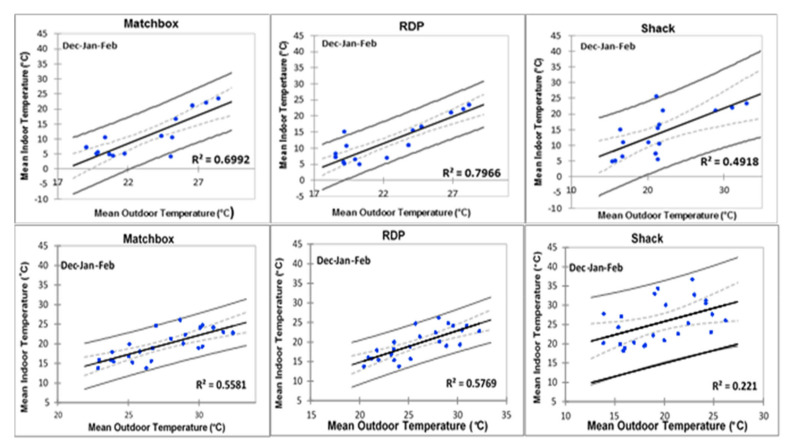
Regression models of three housing types in Kathorus (top row) and Wakkerstroom during warmer months, with a confidence interval of 95% (d.f. = 1).

**Figure 12 ijerph-17-03524-f012:**
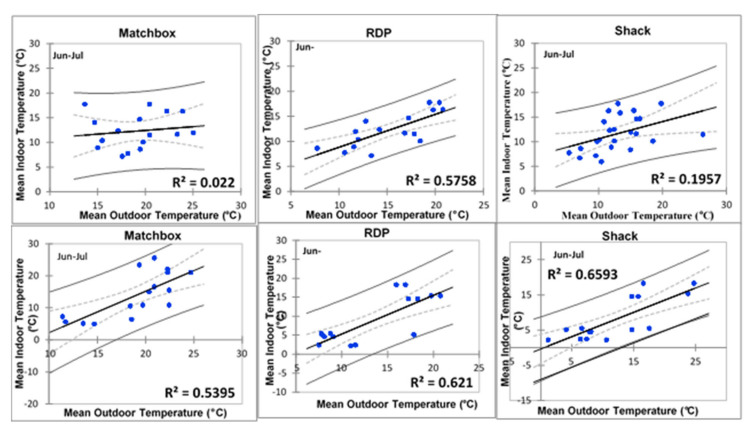
Regression models of the three types in Kathorus (top row) during colder months, with a confidence interval of 95% (d.f. = 1).

**Figure 13 ijerph-17-03524-f013:**
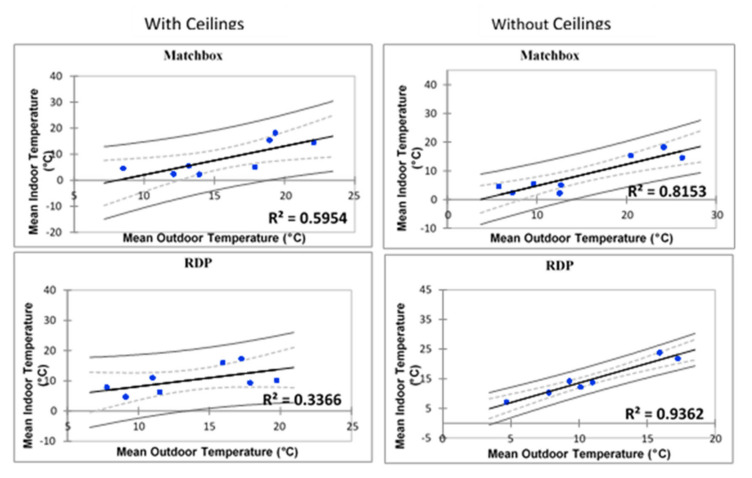
Regression models comparing homes with ceilings (left) and those without (right) of match box and RDP types in Wakkerstroom (with a 95% confidence interval).

**Figure 14 ijerph-17-03524-f014:**
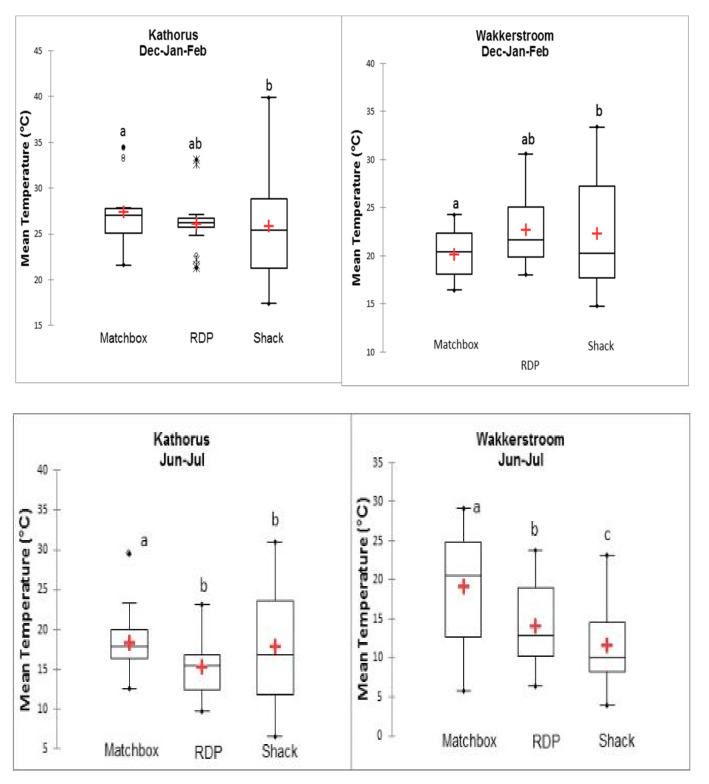
Boxplot comparing daily mean indoor temperature for all three housing types (n = 28) during both warmer and colder months in Kathorus and Wakkerstroom. The top whisker represents the highest value not including outliers; the upper quartile represents that 25% of the data are greater than this value. The middle (median) represents that 50% of the data are greater than this value; the bottom whisker represents that 25% of the data are less than this value; the bottom whisker is the minimum value also without outliers.

**Figure 15 ijerph-17-03524-f015:**
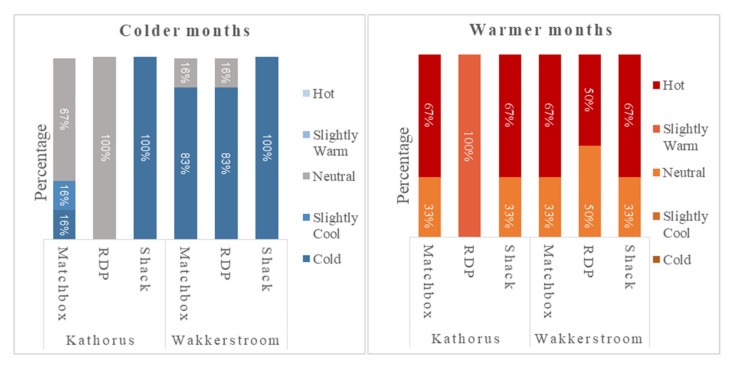
Self-reported perception of thermal comfort during warmer and colder months in Kathorus and Wakkerstroom area (n = 85).

**Table 1 ijerph-17-03524-t001:** Pearson’s correlation of outdoor temperatures vs. indoor temperatures in the Kathorus and Wakkerstroom areas.

		Warmer Months	Colder Months
Kathorus	Matchbox	0.74 (*p* < 0.0001)	0.15 (*p* > 0.05)
	RDP	0.75 (*p* < 0.0001)	0.75 (*p* < 0.0001)
	Shack	0.05 (*p* > 0.05)	0.03 (*p* > 0.05)
Wakkerstroom	Matchbox	0.73 (*p* < 0.0001)	0.6 (*p* < 0.05)
	RDP	0.8 (*p* < 0.0001)	0.75 (*p* < 0.001)
	Shack	0.6 (*p* < 0.05)	0.75 (*p* < 0.001)

**Table 2 ijerph-17-03524-t002:** Percentage of building material used to construct each housing type, as per expert opinion and self-reported accounts. The number of households which used a specific form of building material are quantified.

**Kathorus**	**Housing Type** **(n = 6 per Housing Type)**	**Wall Material**	**Floor Material**	**Ceiling Type**	**Roofing Material**	**Mean Age of Houses**
	Matchbox (Apartheid era)	100% Bricks (low grade clay bricks) (n = 6)	100% (Concrete and Tiles) (n = 4), 100% (Concrete) (n = 2)	-	100% (Corrugated iron) (n = 6)	±60 years
	RDP (Post-1994) Shacks (Informal Housing)	100% Bricks (cement, ash/river sand) (n = 6) 100% Galvanized Iron (80% Corrugated and 20% Other) (n = 6)	100% (Concrete) (n = 6) 100% (Vinyl and Concrete) (n = 6)	- -	100% (Asbestos) (n = 4) 100% (Corrugated iron) (n = 2) 100% Galvanized iron (40% Corrugated and 60% other) (n = 6)	±24 years ±2.5 (Range:1–4) years
**Wakkerstroom**	Matchbox (Apartheid era)	100% Bricks (low grade clay bricks) (n = 6)	100% (Concrete) (n = 5), 100% (Concrete and Tiles) (n = 1)	100% (Rhino-board) (n = 1) 100% (Wood) (n = 1)	100% (Asbestos) (n = 4), 100% (Corrugated iron) (n = 2)	±60 years
	RDP (Post-1994)	100% Bricks (cement, ash and river sand) (n = 6)	100% (Concrete) (n = 6)	100% (Rhino-board) (n = 2)	100% (Corrugated iron) (n = 6)	±21.5 years
	Shack (Informal Housing)	100% Corrugated iron (n = 6)	100% (Vinyl and Concrete)	100% (Cardboard) (n = 1)	100% Galvanized iron (Corrugated iron) (n = 6)	±9.25 (Range: 0.5–18) years

**Table 3 ijerph-17-03524-t003:** Self-reported adaptation methods used in Kathorus and Wakkerstroom during the colder and warmer months.

Housing Type (n = 6 per housing type)		Heating and Cooling of Homes		Clothing Modifications		Behavioral Adaptations		Average Number of Occupants	
		Colder Months	Warmer Months	Colder Months	Warmer Months	Colder Months	Warmer Months	Occupant Number	Range
Kathorus	Matchbox (Apartheid era)	100% Electric Heaters	33% Fans	100% Increasing clothing and add blankets	100% Decreasing clothing and blankets	* 33% Sleep earlier 50% Open windows for circulation	100% Open windows and doors (67%—All day, 33%—2–4 hours)	3 (n = 16)	1–6
	RDP (Post-1994)	100% Electric Heaters	-	50% Increasing clothing and blankets	100% Decreasing clothing and blankets	*50% Sleep earlier	100% Open windows and doors (All day)	3 (n = 20)	2–-5
				50% Increase clothing only		0% opened windows and doors			
	Shack (Informal Housing)	100% Electric Heaters	16 % Fans	67% Increase clothing and blankets	100% Decreasing clothing and blankets	* 67% Sleep earlier	100% Open windows and doors (All day)	2 (n = 13)	1–3
				16% Blankets only		33% Opened windows and doors			
				16% Nothing					
Wakkerstroom	Matchbox (Apartheid era)	67% Wood-fire	33% Fans	100% Increasing clothing and add blankets	100% Decreasing clothing and blankets	* 50% Sleep earlier	100% Open windows and doors (All day)	6 (n = 35)	1–7
		16% Coal fire				67% Open windows for circulation			
	RDP (Post-1994)	50% Wood-fire 30% Electric heater 20% Cow Dung	16% Fans	67% Increase clothing and blankets 16% Blankets only	100% Decreasing clothing and blankets	* 50% Sleep earlier 67% Open windows for circulation	100% Open windows and doors (All day)	5 (n = 30)	2–9
	Shack (Informal Housing)	50% Wood-fire	-	83% Increase clothing and blanket	100% Decreasing clothing and blankets	* 100% Sleep earlier 33% Open windows for circulation	100% Open windows and doors (All day)	3 (n = 15)	1–4
		33% Gas stove		16% Clothing only					
		16% Electric heater		16% Nothing					

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
