# Peer review of "The Three Little Houses: A Comparative Study of Indoor and Ambient Temperatures in Three Low-Cost Housing Types in Gauteng and Mpumalanga, South Africa"

_ijerph, 2020, doi:10.3390/ijerph17103524_

Round 1

Reviewer 1 Report

Comments:

Overall, I feel that the method has to be better explained and the analysis must be revised, amended or complemented. Some general comments are given below.

Abstract: what is the importance of such equipment loss in an abstract?

On the rationale (line 20) Building materials are usually only part of the story, I think typologies and the general state of the dwellings have at least the same degree of importance, plus users' habits.

Lines 44-46 It is really hard to understand this sentence. There is an increase in ambient temperature in South Africa which is twice as much the average increase globally?

Line 50 “ailments”?

Line 50: “weather” you mean, climate?

Line 63: as mentioned before, it goes beyond materials.

Line 69-70: “universal agreed upon standard which has been developed“ which is it? Please introduce it here

Lines 88…: such description could be avoided by presenting floor plans and wall and rood sections as well as building properties

Lines 90-91: “The study did not consider house orientation, which also influences heating and cooling.” How come? Was there a known orientation of each façade or not? Does that mean authors were unaware of their orientation?

Line 114: why the question mark?

Figures 2-7: I appreciate the photos of the homes presented, however the three houses should be better detailed, including a description of wall, floor, roof elements, calculated U-values, thermal mass / heat capacity, occupation etc

Line 134: why 36? Wouldn't it be possible to select one single example of each one for inter-comparisons?

Line 144: There was a previous comment saying that all evaluated houses were at least 15 years old. So why such discrepancies in age? "All formal low-cost 89 houses that were chosen were older than 15 years, in order to minimise building-age-based bias."

On the three housing types: are the three houses comparable? Foor area, typology, orientation... or is this a comparison of mean values for each housing type? Not clear! Please clarify. In the case of grouped data, it is important to inform if variations within samples are less pronounced than between the three housing types.

Line 165: Why at 6 am? Is that cooking time? Internal heat gains?

Figure 8: The mean temperatures are for different houses of for different days?

Figure 11: I understand that the previous analyses grouped matchbox (note a typo in caption) and RDP houses with and without ceiling despite them showing different behavior. Please discuss or amend previous graphs and analyses.

On the relationship indoors/outdoors: OK, but what does it mean the indoor temperatures being more or less related to outdoor temperatures? I miss a discussion and an explanation here for this effect.

On 3.2.1: this analysis is not well resented. what was the sample? how was the survey performed (once, twice, once per season), direct questions, online form?

Line 277: why “interestingly”? Maybe especially because the ceiling provides an extra insulation. Please discuss.

Figure 13: this color scale is not distinctive enough, sometimes is hard to categorize each housing type.

Lines 313-314: maybe you need to check shack results for 6am, as these are the highest indoor temperatures during winter days!

On Discussion: thermal properties are not adequately presented.

The comment on UTCI does not make sense: UTCI is not the same as ambient temperature. It is an index applicable to outdoor conditions and stress categories are not for indoor air temperatures!

Reviewer 2 Report

Review of ijerph-775412-peer-review-v1 The Three Little Houses: A Comparative Study of Indoor and Ambient Temperatures in Three Low-cost 3 Housing Types in Gauteng and Mpumalanga, South Africa

Abstract

What does “semi-structured” interview mean? Usually it either is or is not structured.

The title suggests that this study is about both indoor and outdoor temperatures, but the abstract only talks about indoor temperature fluctuations. What is the relationship between indoor and outdoor temperature? Were any adverse health effects found? Why should fluctuations only be a matter of concern? Is it not more important to examine if dwellings can provide protection from outdoor extremes? Were there any significant differences in the 3 categories of houses? Is not insulation an appropriate recommendation, not only installation of ceilings?

Line 81. What does RDP mean?

Figure 1, the maps are not very informative and could be deleted

Line 188, the p value should not be shown as just p>0.05. Was it close or was it not very close to significance at all?

Line 217 and 218: This sentence is awkward and not clear.

Line 22. It is not sufficient to just report p values > 0.05.

Line 244. The article should say whether or not any insulation materials were present at all.

The discussion section does not include anything about strengths and weaknesses in this study. It does not appear to reference any other similar studies. It should also include a discussion on the importance of documenting adverse health outcomes associated with indoor temperatures, which was a research gap noted in the recently released World Health Organization Guidelines on Housing and Health, which this article should reference. It should be mentioned the WHO recommendation on indoor temperature was “conditional.”  See https://www.who.int/sustainable-development/publications/housing-health-guidelines/en/

Some of these limitations are shown in the conclusions section, but they belong in the discussion section.

The chief weakness of this study was the failure to gather any health information (self reported or otherwise. This might be a recommendation for future research.

The English throughout this article is poor and needs improvement.

End

Reviewer 3 Report

General remarks

Interesting and relevant study. The overall conclusion, that thermal comfort is better inside houses made from concrete or clay bricks compared to sheds made from corrugated iron sheets, is very relevant.

How was the sample size determined? There is a lot of statistics on a relatively low number of houses/thermal loggers.

The pictures could be much better. Many are stretched, others blurred. It would also be usefull to get an impression of the siting and neighborhoods through a map or an aerial photo.

A more systematic description of the buildings would be very useful. What is the floor area, wall height, number of rooms, number of inhabitants?

Air temperature is an indicator of thermal comfort, but it is quite crude as there are lots of other factors, including not least humidity and airspeed. The Ashrea 55 standard (reference 12), is defining methods for combining and evaluating these different environmental factors in combination to predict thermal comfort. Ashrea 55 is also suggesting a standard scale “The ASHRAE thermal sensation scale” for quantifying people’s thermal sensation (+3 hot, +2 warm, +1 slightly warm, 0 neutral,–1 slightly cool,–2 cool, –3 cold)

Specific remarks:

100: It doesn’t seem like a clay brick house. The bricks looks very big more like 60x20 cm.

106: Asbestos roofs without ceiling are a health hazard in itself.

115: The picture of the house on the right: Made from some kind of element? Not bricks.

134: For how long were the houses measured? A year – do they hold data for that long?

135: What is the precision and reliability of the IButtons? And why not measuring every hour?

138: The non-standard position of the loggers inside the houses is a bit tricky. The temperature can vary several degrees in the same room.

Figure 8. It is difficult to understand the overall temperature pattern. Why are the shacks warmest at 6 AM? A building made from corrugated iron has no thermal capacity so it is not so likely to be warmest in the morning, it would normally be warmest a mid-day and afternoon until the sun sets and then cool of quickly. The RDP houses in the cooler months and the matchbox houses in the warmer months are more showings patterns I would expect. Are you sure about the data? It would be good with an additional graph in figure 8 showing outdoor temperatures for comparison.

216: As the heat is radiating to the sky more easily when there is no ceiling

Table 2. “Main construction materials” should be “Wall material”. Roofing material: I suggest only two categories; asbestos and corrugated iron. Floor material: Concrete rather than cement (cement is an ingredient in concrete)

262: It is quite difficult to get reliable data asking people about thermal comfort. That is why Ashrea 55 uses measured data to predict comfort.

  1. It could be because clay bricks are better at absorbing moisture therefore giving a more stable indoor environment.
  2. Plaster is not really an insulating material. Is it because the plaster closes the cracks? Plaster has thermal properties similar to concrete.

Round 2

Reviewer 1 Report

Authors have revised the manuscript quite thoroughly. I recommend its publication.

Reviewer 2 Report

this revision is ok, still some minor English errors